

# Alkalinity and nitrate concentrations in calcareous watersheds: Are they linked, and is there an upper limit to alkalinity?

Beat Müller[1], Joseph S. Meyer[2,3], René Gächter[1]

[1]Eawag, Swiss Federal Institute of Aquatic Science and Technology, CH-6047 Kastanienbaum, Switzerland
[2]Department of Chemistry and Geochemistry, Colorado School of Mines, Golden, Colorado 80401 USA
[3]Applied Limnology Professionals LLC, Golden, Colorado 80401 USA

*Correspondence to:* Beat Müller (beat.mueller@eawag.ch)

**Abstract.** Data from aquifers in calcareous watersheds in Switzerland demonstrate that alkalinity initially increases approximately in proportion to nitrate ($NO_3^-$) concentration in the groundwater and eventually approaches an apparent
maximum of approximately 8 mmol $L^{-1}$ at high $NO_3^-$ concentrations. This close positive relationship between alkalinity and $NO_3^-$ concentration appears to be predominantly a result of three processes: (i) mineralization of organic nitrogen (N) fertilizer, (ii) exchange of $OH^-$ and $H^+$ during the uptake of $NO_3^-$ or ammonium ($NH_4^+$), and (iii) $CO_2$ released by roots as a result of fertilizer-stimulated plant growth. Atmospheric deposition of N and strong acids ($H_2SO_4$ and $HNO_3$) play a minor role. We suggest that the asymptotic approach to a maximum groundwater alkalinity at $NO_3^-$ concentrations exceeding 0.25
mmol $L^{-1}$ may be caused by (i) a maximum possible areal crop production at excessive N fertilization and (ii) an increasing $CO_2$ loss to the atmosphere due to the increasing $CO_2$ production in the soil. Thus, we estimate that the fertilizer-intensive agriculture of Switzerland generates an annual flux from the soil to the atmosphere of at least 0.26 Mt $CO_2$ $a^{-1}$. This analysis provides a general understanding and quantitative prediction of steady-state groundwater $NO_3^-$ concentration; equilibrium groundwater alkalinity, pH, and $pCO_2$; and soil $CO_2$ emissions to the atmosphere based on quantitative and qualitative
information on the supply of N and acidity to the soil by atmospheric deposition and N fertilization. The positive correlation between alkalinity and $NO_3^-$ concentration in groundwaters persists in rivers and lakes. However, due to the diffusive loss of $CO_2$ to the atmosphere, subsequent precipitation of calcite, dilution with surface water, input of wastewater discharges and $NO_3^-$ consumption by aquatic photoautotrophs, the correlation is less distinct.

## 1 Introduction

To enhance crop production, farmers supply nitrogen (N) in the form of inorganic and organic fertilizers (Walworth, 2013) and/or manure to fields. Because nitrification [the oxidation of ammonium ($NH_4^+$) to nitrate ($NO_3^-$)] of reduced-N compounds produces protons ($H^+$) that can dissolve carbonate minerals (Equations S-1 to S-5 and S-13 to S-17 in SI Table S1), the effects of N-fertilizers on dissolved inorganic carbon (DIC) concentrations and fluxes in lotic systems have been investigated (Semhi et al., 2000; Raymond and Cole, 2003; Raymond et al., 2008; Barnes and Raymond, 2009; Kaushal et
al., 2013; Müller et al., 2016). Additionally, the potential for a resulting release of $CO_2$ to the atmosphere and its consequences for global climate change have been of special concern (West and McBride, 2005; Perrin et al., 2008; Li et al., 2013).

Alkalinity is a measure of the acid-neutralizing capacity of water. In freshwater systems, it is generally defined as:

$$\text{Alkalinity (as mmol } L^{-1}) = \left[HCO_3^-\right] + 2\left[CO_3^{2-}\right] + \left[OH^-\right] - [H^+] \tag{1}$$

where the brackets indicate concentrations of the designated ions (Stumm and Morgan, 1996). At circumneutral pH, bicarbonate ion ($HCO_3^-$) is the major component of alkalinity, with only minor contributions from carbonate ($CO_3^{2-}$) and





hydroxyl ($OH^-$) ions. In calcareous watersheds, alkalinity originates mainly from the dissolution of carbonate minerals [e.g., calcite ($CaCO_3$) and dolomite ($Ca_xMg_{(1-x)}CO_3$)] either by i) protons, which are generated by dissociation of strong mineral acids [e.g., nitric ($HNO_3$) or sulfuric ($H_2SO_4$)], or ii) by carbonic acid ($H_2CO_3$) that is formed by the solvation of biogenic $CO_2$ in water (Equations 2 and 3).

$$CaCO_3 + H^+ = Ca^{2+} + HCO_3^- \qquad (2)$$

$$CaCO_3 + H_2CO_3 = Ca^{2+} + 2HCO_3^- \qquad (3)$$

Therefore, in calcareous soils, in-soil production of $CO_2$ (e.g., due to root respiration and heterotrophic mineralization of organic matter) and/or generation of protons by nitrification result in an increased alkalinity concentration. In contrast, in the absence of carbonate minerals, alkalinity is expected to decrease in proportion to the amount of nitrification and in-soil $CO_2$ production (Perrin et al., 2008).

Changes in alkalinity can be of importance to the global carbon budget. For example, the alkalinity export of the Mississippi River (USA) to the Gulf of Mexico increased between 1950 and 2000, in excess of the simultaneously increasing water discharge (Raymond and Cole, 2003). West and McBride (2005) estimated 38% of the $HCO_3^-$ transported by the Mississippi River originated from calcite dissolution induced by $HNO_3$. Similarly, Semhi et al. (2000), Perrin et al. (2008), Probst (1986) and Brunet et al. (2011) have attributed increasing alkalinity observed in French rivers to nitrification of N-fertilizers, and Guo et al. (2010) observed increased soil acidity as a consequence of the increasing use of N-fertilizers in China.

Apart from the increasing partial pressure of $CO_2$ ($pCO_2$) in the atmosphere (NOAA, 2017), increasing soil temperature (Zobrist et al., 2018) and changes in runoff, a variety of factors may affect groundwater and thus riverine alkalinity (Raymond et al., 2008). Those factors include (i) agricultural practices such as fertilization and liming (Müller et al., 2016; West and McBride, 2005; Oh and Raymond, 2006), (ii) mineral-acid load originating from atmospheric deposition (e.g., $HNO_3$ [Equation S-6 in SI Table S1] and $H_2SO_4$), (iii) changes in microbial activity, (iv) acid mine drainage (Kaushal et al., 2013), and (v) sewage effluents (Barnes and Raymond, 2009). Because N plays a leading role in many of these interactions with soil water, groundwater and surface waters globally, and in view of the approximately five-fold increase of anthropogenic reactive N since 1960, with uncertain consequences, Battye et al. (2017) posed the question "Is nitrogen the next carbon?".

In this study, we

- present data from extensive groundwater and surface-water monitoring programs in calcareous agricultural Swiss watersheds demonstrating a strong linkage between alkalinity and $NO_3^-$ concentrations,
- identify agricultural N fertilization fueling terrestrial primary production as the main driving force linking the two constituents in ground and surface waters.
- explain why alkalinity reaches an apparent upper limit at elevated $NO_3^-$ concentrations in groundwater, and
- estimate the contributions of the Swiss farming activities to the global $CO_2$ accumulation and to the alkalinity load of downstream surface waters.

## 2 Methods

Aquifer and surface waters considered in this study are described in Table 1. At atmospheric $pCO_2$ ranging from $10^{-3.5}$ to $10^{-3.4}$ atm (the condition from approximately 1959 to present; NOAA, 2017) and in the absence of acids other than $H_2CO_3$, an alkalinity concentration of 1.42 mmol $L^{-1}$ and a pH of 8.24 would be expected in water draining a hypothetically N-free (and thus sterile) calcareous soil, assuming a groundwater temperature of 8 °C. Therefore, to avoid potentially confounding





results from waters not in considerable contact with carbonates, we included in this analysis only waters originating from calcareous watersheds in which alkalinity exceeded 1 mmol $L^{-1}$. If a titrated alkalinity was not reported, we approximated it as the reported molar concentration of $HCO_3^-$.

In SI Table S1, we calculated theoretical molar $\Delta[HCO_3^-]:\Delta[NO_3^-]$ and $\Delta([Ca]+[Mg]):\Delta[HCO_3^-]$ ratios of chemical reactions

that likely occur in aquifers in the presence or absence of carbonate minerals. Additionally, we applied the geochemical speciation software ChemEQL Version 3.2 (Müller, 2015) to calculate pH, alkalinity, $pCO_2$, and $Ca^{2+}$ concentrations in groundwater in equilibrium with calcite, into which we computationally titrated the $CO_2$, $NO_3^-$, and $H^+$ resulting from the estimated loads of these compounds as used in Swiss agriculture. For these computational titrations, we neglected the exchange of soil/groundwater $CO_2$ with the atmosphere and assumed equilibrium of the groundwater with soil $pCO_2$. Initial

conditions were as follows: temperature = 8 °C, atmospheric $pCO_2$ = 0.0004 atm in equilibrium with calcite, and $NO_3^-$ = 0 mmol/L, resulting in pH = 8.24 and alkalinity = 1.42 mmol $L^{-1}$.

We calculated the $CO_2$ saturation index of water as $\Omega_{CO_2}$ = $CO_2(aq)/CO_2(atm)$, where $CO_2(aq)$ is the partial pressure of $CO_2$ in the water (in atm) and $CO_2(atm)$ is the partial pressure of $CO_2$ in the atmosphere (in atm). $CO_2(aq)$ was calculated from the reported pH and alkalinity using Equation 6 in Müller et al. (2016). Reported concentrations of $Ca^{2+}$, alkalinity, and $NO_3^-$

in lakes represent values observed at spring overturn averaged over all years of observation. Concentrations measured in rivers were averaged over the observation period. To produce conservative (i.e., minimum) estimates of $\Omega_{CO_2}$, we assumed a value of $4x10^{-4}$ atm (400 ppm) for $CO_2(atm)$, which is the 2015 annual average in the Mauna Loa dataset (NOAA, 2017). The saturation index for calcite solubility was calculated as

$$\Omega_{calcite} = \frac{\{Ca_2^+\} \times \{CO_3^{2-}\}}{K_{calcite}} \qquad (4)$$

where braces denote the ChemEQL-predicted chemical activity of $Ca^{2+}$ or $CO_3^{2-}$, respectively, and $K_{calcite}$ is the solubility product of calcite ($3.95 \times 10^{-9}$ at 8 °C for groundwaters; $3.71 \times 10^{-9}$ at 15 °C for rivers). Ion activities were estimated with the Debye-Hückel approximation (Stumm and Morgan, 1996) for an ionic strength calculated from the sum of measured anions

and cations. We calculated $\Omega_{calcite}$ only for the NAQUA-groundwaters and rivers, because $Ca^{2+}$ concentrations were not available for lake waters and Zürich well waters.

Linear regressions of alkalinity *versus* $NO_3^-$ concentration were performed with the statistical software program Stata Version 14 (StataCorp LLC, College Station, Texas, USA). Regression coefficients were inferred to be statistically significant when $p \leq 0.05$.


### 3 Results

Alkalinity increased systematically with increasing $NO_3^-$ concentrations in groundwater as well as in surface waters (Figures 1a and 1b). In the concentration range 0-0.25 mmol $NO_3^-$ $L^{-1}$, initial slopes of alkalinity *versus* $NO_3^-$ concentration for groundwater and surface waters were 17.0 and 12.5 mmol alkalinity $L^{-1}$ / mmol $NO_3^-$ $L^{-1}$, respectively (Table 2). The

alkalinity-axis intercepts for the aquifer and surface waters were 2.44 and 1.74 mmol $L^{-1}$, respectively (Table 2). At $NO_3^-$ concentrations exceeding 0.25 mmol $L^{-1}$, the alkalinity *versus* $NO_3^-$ slope decreased in the aquifer waters nearly 10-fold to only 1.8 mmol alkalinity $L^{-1}$ / mmol $NO_3^-$ $L^{-1}$ (Table 2, Figure 1a). Because $NO_3^-$ concentrations in the surface waters did not exceed 0.25 mmol $L^{-1}$ (Figure 1b), we could not test for an analogous decrease of the slope at elevated $NO_3^-$ concentrations in those waters.



All waters were distinctly supersaturated with respect to atmospheric $CO_2$ (up to 90-fold in groundwaters and up to 11-fold in surface waters; Figure 1c). In the groundwaters, $\Omega_{CO_2}$ increased rapidly as $NO_3^-$ concentration increased from 0 to approximately 0.25 mmol L$^{-1}$, but leveled-off at higher $NO_3^-$ concentrations. In the surface waters, the trend of increasing $\Omega_{CO_2}$ with increasing $NO_3^-$ concentration was less distinct.

As $NO_3^-$ concentration increased from 0 to 0.25 mmol L$^{-1}$ in the groundwaters, pH tended to initially decrease but then leveled-off to approximately circumneutral values at higher $NO_3^-$ concentrations (Figure 1d). In surface waters, pH ranged between 7.7 and 8.3, seemingly unrelated to the $NO_3^-$ concentration (Figure 1d).

Ranging from 0.51 to 0.72, the molar ([Ca]+[Mg]) : [HCO$_3^-$] ratio in the aquifer waters increased significantly as $NO_3^-$ concentration increased (slope = 0.079 / mmol $NO_3^-$ L$^{-1}$, p < 0.001; Figure 1e). However, $NO_3^-$ only accounted for 15% of

the variation of this ratio, suggesting that the majority of that variation depended on other factors. In the surface waters, the molar ([Ca]+[Mg]) : [HCO$_3^-$] ratio ranged from 0.47 to 0.74, without a trend analogous to the groundwater trend across the narrower range of $NO_3^-$ concentrations in the surface waters (Figure 1f).

Contrary to most groundwaters being slightly undersaturated with respect to $CaCO_3$ solubility ($\Omega_{CaCO_3}$ = 0.5-1.2; Figure 1g), 21 of the 22 river waters were supersaturated ($\Omega_{CaCO_3}$ = 1.2-8.4; Figure 1g).

Table 3 summarizes quantitative information about the various sources, sinks and pathways affecting the nitrogen budget of agricultural Swiss soils.

## 4 Discussion

### 4.1 Aquifer Waters

Because in all aquifer waters included in this analysis, (i) alkalinity exceeded the 1.42 mmol L$^{-1}$ concentration expected in
calcite-saturated, air-equilibrated water (Figure 1a), (ii) pH was considerably below the equilibrium-predicted value of 8.24 (Figure 1d), and (iii) $CO_2$ concentrations were highly supersaturated (Figure 1c), the aquifer waters appear to be at least partially protected from contact with the atmosphere.

The equations in Table S1 show that the stoichiometric molar ([Ca]+[Mg]) : [HCO$_3^-$] ratio (i) equals 0.5 if carbonates are dissolved only as a result of interaction with $CO_2$ (Equations S-19 and S-20), (ii) exceeds 0.5 if protons generated by
nitrification or supplied by acid precipitation (Equations S-1 to S-6) contribute to the dissolution of carbonates (Equations S-13 to S-18), and (iii) only slightly exceeds 0.5 (Equation S-21) if mineralization of organic matter (e.g., $C_{106}H_{263}O_{110}N_{16}P$, per Stumm and Morgan, 1996) is responsible for the carbonate dissolution. In all the groundwaters, the ([Ca]+[Mg]) : [HCO$_3^-$] ratio slightly exceeded 0.5 (Figure 1e), suggesting that protons generated by in-soil nitrification or atmospheric deposition of acids must have contributed partly but not extensively to the generation of alkalinity in the groundwaters.

Despite high $pCO_2$ in the groundwaters, those waters were only saturated to slightly undersaturated with respect to calcite solubility (Figure 1g), likely due to slow, diffusion-limited kinetics of calcite dissolution in the relatively quiescent soil water (Plummer et al. 1979).

#### 4.1.1 Origin of nitrogen in Swiss groundwaters

Swiss legislation (Art. 14 Abs. 4 GSchG) restricts fertilization of crop fields and meadows to a maximum of 315 kg N ha$^{-1}$ a$^{-1}$ (22.5 kmol N ha$^{-1}$ a$^{-1}$). Table 3 and Figure 2 display that on average the total N-fertilizer supply in Switzerland amounts to 206 kg N ha$^{-1}$ a$^{-1}$, seed adds 1 kg N ha$^{-1}$ a$^{-1}$, and atmospheric deposition contributes 26 kg N ha$^{-1}$ a$^{-1}$. On the other hand, only 148 kg N ha$^{-1}$ a$^{-1}$ are harvested as crops, resulting in an annual surplus of 85 kg N ha$^{-1}$ a$^{-1}$ to the soil. Because N does not accumulate in soils, this excess supply must result in losses to the groundwater (22 kg N ha$^{-1}$ a$^{-1}$) and to the atmosphere (63

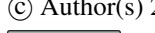


kg N ha$^{-1}$ a$^{-1}$) by way of denitrification to N$_2$ and/or N$_2$O (Table 3). These inputs via fertilizer and atmospheric deposition and losses via crops and denitrification are summarized in Figure 2 (green arrows).

Because the annual groundwater generation rate is approximately 5000 m$^3$ ha$^{-1}$ a$^{-1}$, we estimate an average groundwater NO$_3^-$ concentration of 22 kg N/5000 m$^3$ a$^{-1}$ = 4.4 mg N L$^{-1}$ or 0.31 mmol N L$^{-1}$. This value is in the mid-range of the observed

NO$_3^-$ concentrations in Figure 1.

### 4.1.2 CO$_2$ and acidity production by organic nitrogen fertilization

As indicated in Table 3, in Switzerland most of the N-fertilizer is applied as manure (124 kg ha$^{-1}$ a$^{-1}$) or by way of N fixation by Leguminosae (32 kg ha$^{-1}$ a$^{-1}$). Because approximately 20% of the 50 kg ha$^{-1}$ a$^{-1}$ of commercial N-fertilizer (i.e., 10 kg ha$^{-1}$ a$^{-1}$) is supplied as urea (FOAG, 2016), a total of 166 kg N ha$^{-1}$ a$^{-1}$ or 11.9 kmol N ha$^{-1}$ a$^{-1}$ of organic N needs to be

mineralized annually before becoming bioavailable to plants. According to Equations S-5 and S-9 in SI Table S1, this mineralization produces totals of 11.9 kmol NO$_3^-$ ha$^{-1}$ a$^{-1}$, 74.2 kmol CO$_2$ ha$^{-1}$ a$^{-1}$ and 13.3 kmol H$^+$ ha$^{-1}$ a$^{-1}$ (Fig. 2).

### 4.1.3 Nitrate supply and acidity production due to the application of inorganic nitrogen fertilizer

If the remaining 40 kg N ha$^{-1}$ a$^{-1}$ (2.86 kmol N ha$^{-1}$ a$^{-1}$) of commercial fertilizer is applied as NH$_4$NO$_3$, its nitrification results in production of 2.86 kmol H$^+$ ha$^{-1}$ a$^{-1}$ and 2.86 kmol NO$_3^-$ ha$^{-1}$ a$^{-1}$ (Equation S-2 in SI Table S1, and Figure 2).

### 4.1.4 Contribution of atmospheric nitrogen, acid and CO$_2$ deposition to alkalinity production

Strong acids in the atmosphere originate from combustion of fuels that generates NO$_x$ and SO$_2$, which in contact with atmospheric water droplets oxidize to nitric and sulfuric acids. The average atmospheric deposition of acidity in Switzerland (Swiss plateau, Jura and pre-alps) is estimated to be 1.34 kmol H$^+$ ha$^{-1}$ a$^{-1}$ (Graf Pannatier et al., 2011, their Table 3).

In Switzerland, approximately one-third of the 26 kg N ha$^{-1}$ a$^{-1}$ (1.86 kmol N ha$^{-1}$ a$^{-1}$) of atmospheric deposited N consists of

NO$_3^-$ and two-thirds of NH$_4$OH (NABEL, 2016). Nitrification of the latter in aerated soil or groundwater generates 1.24 kmol acidity ha$^{-1}$ a$^{-1}$ (Equation S-1). Thus, the atmospheric acid deposition results in a total acidity load of 1.34 + 1.24 = 2.6 kmol H$^+$ ha$^{-1}$ a$^{-1}$.

At a pCO$_2$ of 4x10$^{-4}$ atm, approximately 10$^4$ m$^3$ ha$^{-1}$ a$^{-1}$ of rain water (FOMC, 2018) saturated with CO$_2$ imports 0.26 kmol CO$_2$ ha$^{-1}$ a$^{-1}$ into the soil.

### 4.1.5 Nitrate and acidity removal, and CO$_2$ generation due to crop production

In the process of NO$_3^-$ assimilation, plants take up protons from the soil water in order to maintain charge balance (Haynes, 1990):

$$106CO_2 + 16\,NO_3^- + HPO_4^{2-} + 122H_2O + 18H^+ \Leftrightarrow \{C_{106}H_{263}O_{110}N_{16}P\} + 138O_2 \qquad (5)$$

where C$_{106}$H$_{263}$O$_{110}$N$_{16}$P is a generic formula for organic matter (eg. Stumm and Morgan, 1996). Accordingly, an annual net

primary production of 148 kg N ha$^{-1}$ (10.6 kmol N ha$^{-1}$ a$^{-1}$) removes 11.9 kmol acidity ha$^{-1}$ a$^{-1}$ from the soil.

In addition, growth-dependent root respiration of crop plants can contribute 12-62% of the total in-soil CO$_2$ production in temperate-zone crop fields, grasslands, and forests (Raich and Tufekcioglu, 2000). Using sunflower as a model plant, root respiration amounts to approximately 20% of net assimilation (Cheng et al., 2000). Assuming a molar C:N ratio of 106:16, the net assimilation of 10.6 kmol N ha$^{-1}$ a$^{-1}$ corresponds to a net carbon production of 70.2 kmol C ha$^{-1}$ a$^{-1}$ and thus suggests

an in-soil CO$_2$ production by crop plants of approximately 14 kmol CO$_2$ ha$^{-1}$ a$^{-1}$.

### 4.1.6 Alkalinity and nitrate concentrations as well as CO$_2$ emissions from the soil to the atmosphere are directly related to agricultural production



Growth-dependent N uptake by crop plants increases with increasing N fertilization but eventually reaches a maximum at elevated concentrations in the soil solution (Baule, 1917). Mineralization of increasing amounts of manure stimulates growth-dependent N uptake that results in increased groundwater $NO_3^-$ and alkalinity concentrations, mainly due to the coupling of plant growth and root respiration. But as expected from the concept of a maximum of growth-dependent N

uptake and as empirically demonstrated in Figure 1a, alkalinity asymptotically approaches a maximum at elevated $NO_3^-$ concentrations. This result suggests that crop production is no longer limited by N when groundwater concentrations exceed approximately 0.25 mmol $NO_3^-$ $L^{-1}$. The tolerance limit for drinking water (0.4 and 0.7 mmol $NO_3^-$ $L^{-1}$ in Switzerland and USA, respectively) exceeds this value by approximately two- to three-fold. Accordingly, ecologically- and economically-efficient agricultural practice avoiding over-fertilization with N (i.e., not exceeding 0.25 mmol $NO_3^-$ $L^{-1}$) in no way is a risk

for a groundwater drinking-water supply.

Apart from these biological processes, the increase of $CO_2$ oversaturation (Figure 1c) and the accompanying decrease in pH (Figure 1d) as $NO_3^-$ concentration increases suggest that at least two physical-chemical processes might also contribute to the leveling-off of alkalinity at high $NO_3^-$ concentrations. First, in a partly closed groundwater system, not all of the biogenic $CO_2$ and protons are consumed by carbonate-dissolution reactions. As $pCO_2$ increases and as pH concurrently decreases, the

extent to which $H_2CO_3$ dissociates into $HCO_3^-$ and $CO_3^{2-}$ decreases (because increasingly greater percentages of the DIC remain as $H_2CO_3$ as the $H^+$ concentration increases). Thereby, alkalinity increases at a decreasing rate as $pCO_2$ increases. Second, as $pCO_2$ increases in soil water, the rate of diffusive losses of $CO_2$ through air channels in the soil increases. That tendency to diffusively lose $CO_2$ to the atmosphere at increasingly faster rates slows the rate of alkalinity increase, as $NO_3^-$ concentration and associated biogenic $CO_2$ production increase.


### 4.1.7 Synthesis

According to the compilation shown in Figure 2 for Switzerland, fertilization (including atmospheric deposition) of the soil contaminates groundwater with 1.52 kmol N $ha^{-1}$ $a^{-1}$. This results in an average groundwater $NO_3^-$ concentration of 0.30 mmol $L^{-1}$ and in a net-production of 2.36 kmol $H^+$ $ha^{-1}$ $a^{-1}$ and 94.1 kmol $CO_2$ $ha^{-1}$ $a^{-1}$. Subsequent equilibration of the 2.36

kmol $H^+$ and 94.1 kmol $CO_2$ in 5000 $m^3$ water in contact with solid $CaCO_3$ results in pH 7.14, 5.56 mmol $L^{-1}$ alkalinity and a $pCO_2$ of 0.019 atm (1.1 mmol $L^{-1}$). These estimates (light blue circles in Figures 1a, c, d and e) agree well with the actual groundwater measurements.

Alternatively, assuming a scenario in which i) fields are neither fertilized nor harvested, ii) atmospheric deposition (currently 1.86 kmol N $ha^{-1}$ $a^{-1}$) is the only N source, and iii) denitrification removes the same fraction of N as in the fertilized scenario

(74%), the resulting $NO_3^-$ loss via exfiltrating groundwater would be 0.48 kmol N $ha^{-1}$ $a^{-1}$ (i.e., a groundwater concentration of 0.097 mmol $NO_3^-$ $L^{-1}$). This value is 3.1 times lower than the fertilization scenario and is in the mid-rage of the minimum $NO_3^-$ concentrations measured in Swiss groundwaters (Figure 1a). Assuming that the resulting production of $CO_2$ and acidity as well as denitrification in the soil would be 3.1-fold lower than shown in Figure 2, the original groundwater concentrations would be 4.58 mmol $CO_2$ $L^{-1}$ and 0.23 mmol acidity $L^{-1}$ before reaction with soil carbonates. Equilibration with the soil

carbonates would then result in an alkalinity of 3.30 mmol $L^{-1}$, a pH of 7.54, a $\Omega_{CO_2}$ of 11.5 and a $[Ca^{+2}]/[HCO_3^-]$ ratio of 0.54. These values, shown as light blue squares in Figures 1a, c, d, and e, also agree well with the observed data.

Finally, our dataset allows to assess the effect of Swiss farming practice on the atmospheric $CO_2$ budget (Figure 3). As shown in Figure 2 (blue arrows), the soil is exposed to a total annual $CO_2$ load of 94.1 kmol $CO_2$ $ha^{-1}$ $a^{-1}$. Because $CO_2$ neither accumulates infinitely nor is assimilated in soils, it must leave the aquifer as gaseous $CO_2$ or dissolved in the

exfiltrating groundwater either as $H_2CO_3^*$ (= $H_2CO_3$ + $CO_2$(aqueous)) or as dissolved $Ca(HCO_3)_2$. In the 47-fold supersaturated groundwater, the $H_2CO_3^*$ concentration equals 1.1 mol $m^{-3}$. Multiplication with the volume of the annually exfiltrating groundwater (5000 $m^3$ $ha^{-1}$ $a^{-1}$) results in a $H_2CO_3^*$ export of 5.5 kmol $ha^{-1}$ $a^{-1}$.





In addition, 27.8 kmol ha$^{-1}$ a$^{-1}$ of alkalinity is exported with the groundwater. This is partly due to the 2.36 kmol H$^+$ ha$^{-1}$ a$^{-1}$ introduced to the system, which generates an equal amount of alkalinity by dissolution of carbonates (Equation 2). However, the remaining 25.44 kmol ha$^{-1}$ a$^{-1}$ of exported alkalinity must have resulted from the reaction of 12.72 kmol ha$^{-1}$ a$^{-1}$ of CO$_2$ (predominantly present as H$_2$CO$_3$ in the water) with carbonates (Equation 3). Subtracting the sum of these two CO$_2$ losses

via groundwater (5.5+12.72=18.22 kmol ha$^{-1}$ a$^{-1}$) from the total CO$_2$ exposure (94.1 kmol CO$_2$ ha$^{-1}$ a$^{-1}$) results in a gaseous CO$_2$ loss to the atmosphere of 75.9 kmol CO$_2$ ha$^{-1}$ a$^{-1}$.

On the other hand, crop production extracts 10.6 kmol N ha$^{-1}$ a$^{-1}$ from the soil, corresponding to a 70.2 kmol CO$_2$ ha$^{-1}$ a$^{-1}$ net consumption from the atmosphere. Consequently, Swiss agricultural activities burden the atmosphere annually with 5.7 kmol CO$_2$ ha$^{-1}$ a$^{-1}$. Extrapolation to the total Swiss agricultural area yields an annual accumulation of 0.26 Mt CO$_2$ a$^{-1}$. This is a

very conservative estimate, because it neglects the CO$_2$ emissions from CO$_2$-supersaturated surface waters and does not consider that a large fraction of the harvest serves as food for animals, which generate additional CO$_2$.

### 4.2. Surface Waters

Although groundwater exfiltrates to riverbeds or lakes, surface-water quality is in addition subjected to surface runoff and wastewater discharges.

When CO$_2$-supersaturated groundwater exfiltrates and becomes surface water, much of the above-saturation CO$_2$ diffuses into the atmosphere. Additionally, photoautotrophic organisms may assimilate some of the CO$_2$. Both processes lead to the observed higher pH and lower $\Omega_{CO_2}$ values in rivers and lakes than in groundwaters at similar NO$_3^-$ concentrations (Figures 1c and d). Nevertheless, CO$_2$ concentrations in the river and lake waters did not reach equilibrium with the atmospheric pCO$_2$, possibly because of heterotrophic mineralization and slow diffusional loss of the CO$_2$ to the atmosphere (Hotchkiss et

al., 2015).

Calcite would be expected to precipitate due to the loss of CO$_2$ and the rising pH when groundwater exfiltrates. However, as demonstrated in Figure 1g, it remains oversaturated in surface waters, indicating that precipitation of calcite often is kinetically constrained (Plummer et al., 1979) and/or at least partly inhibited (e.g., by elevated phosphate concentrations; Langmuir, 1997: p. 222; Müller et al., 2016).

Wastewater loads to a river may alter its groundwater-derived alkalinity-NO$_3^-$ relationship in several ways. First, introduction of NH$_4^+$-containing wastewater into well-oxygenated river water results in nitrification that increases the NO$_3^-$ concentration but simultaneously decreases the pH and alkalinity if carbonate minerals are not present (Equation S-1 in SI Table S1) or do not dissolve fast enough, thus contributing to an additional release of CO$_2$ to the atmosphere. Second, the introduction of large amounts of wastewater-derived dissolved organic matter stimulates microbial respiration, resulting in

low O$_2$ concentrations that favor denitrification and/or dissimilatory NO$_3^-$ reduction to ammonia, both of which decrease NO$_3^-$ concentrations but increase alkalinity even in the absence of carbonate minerals (Equations S-10 to S-12 in SI Table S1).

Consequently, as demonstrated by Abril et al. (2000) and Abril and Frankignoulle (2001) in the highly-polluted Scheldt Basin (Belgium and Netherlands), large wastewater loads can induce N transformations in river water and change its acid-

base properties to the extent that the positive relationship between alkalinity and NO$_3^-$ concentration observed in calcareous soils completely reverses in a river. Although that reversal did not occur in the Swiss surface waters, a trend toward lower alkalinity at a given NO$_3^-$ concentration is apparent (Figure 1b).





## 5 Conclusions

As illustrated in Figure 2 and discussed above, N-limited terrestrial primary production plays a dominant role in the linking of alkalinity and $NO_3^-$ concentrations in aquifer and surface waters. From a global perspective, application of N-fertilizers fueling the production of short-lived organic matter contributes to a net increase of atmospheric $CO_2$ concentrations via

5    release of $CO_2$ that otherwise would remain sequestered in carbonate minerals in soils (West and McBride, 2005; Perrin et al., 2008; Li et al., 2013), and accelerates the translocation of DIC from calcareous soils to downstream reaches in surface waterways and eventually to the oceans.

As atmospheric $CO_2$ concentrations increase and thus riverine, lake, and oceanic pH values decrease through time, the proportion of the DIC that precipitates as $CaCO_3$ after $CO_2$-oversaturated aquifer waters ex-filtrate to surface-water bodies

10    will decrease, and the proportion that is released to the atmosphere as $CO_2$ will increase. Therefore, continued N-fertilization will contribute to the ongoing global $CO_2$ enrichment of the atmosphere and will promote in-soil calcite dissolution.



**Information about the Supplement**

Table S-1 lists chemical reactions for the nitrification of some common N-fertilizers and several other processes in groundwaters and surface waters in the presence and absence of carbonate minerals. Table S-2 reports results of regressions of alkalinity *versus* $NO_3^-$ concentration in Swiss groundwaters, Canton Zürich well waters, and Swiss lakes and rivers.

*Author contributions.* BM, JSM, and RG contributed equally to the manuscript. BM performed ChemEQL speciation calculations.

*Competing interests.* The authors declare that they have no conflict of interest.

*Acknowledgements*. Financial support was provided by Eawag (Switzerland) and Applied Limnology Professionals LLC. Groundwater data from the Canton of Zürich were provided by the Office of Waste, Water, Energy and Air (WWEA) of the Canton of Zürich, Switzerland. Data of the National Groundwater Monitoring program (NAQUA), module 'TREND', were provided by the Swiss Federal Office for the Environment (FOEN).



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





**Table 1:** Data sources.

| Dataset | Description | Range of NO₃⁻ (mmol L⁻¹) | Range of alkalinity (mmol L⁻¹) |
|---|---|---|---|
| Aquifers | Groundwaters and well waters in carbonaceous watersheds in Switzerland with alkalinity $\geq 1$ mmol L$^{-1}$. Groundwater data from 42 watersheds during 1997-2016 (NAQUA, 2017); averages were calculated over all years. Well-water data from 120 wells along rivers in the Canton of Zürich during 1991-2016 (AWEL, 2017); averages were calculated over all years. | 0.004 - 0.905 | 2.08 - 7.88 |
| Surface waters | Lake and river waters in carbonaceous watersheds in Switzerland with alkalinity $\geq 1$ mmol L$^{-1}$. Lake data from 21 lakes (0.2-580) km$^2$ during 1980-2015 (BAFU, 2017); based on volume-weighted mean concentrations at winter/spring lake overturn averaged over all years. Only 16 of the lakes had Ca and Mg data. River data from 22 stations on 12 rivers during 5- to 44-year monitoring periods (Eawag, 2017); based on concentrations at lake overturn averaged across all years. | 0.017 - 0.219 | 1.52 - 5.03 |



**Table 2:** Regressions of alkalinity *versus* nitrate ($NO_3^-$) concentration for the Swiss aquifer and surface-water data in Figure 1.[a]

| Waters | $NO_3^-$ range for regression (mmol L$^{-1}$) | Slope Value (mmol Alk/ mmol $NO_3^-$) | p | Intercept Value (mmol L$^{-1}$) | p | Regression $R^2$ | n |
|---|---|---|---|---|---|---|---|
| Aquifers | <0.25 | 17.04 (13.89 - 20.19) | <0.001 | 2.44 (1.95 - 2.93) | <0.001 | 0.681 | 57 |
| | >0.25 | 1.79 (0.72 - 2.86) | 0.001 | 5.62 (5.15 - 6.09) | <0.001 | 0.097 | 105 |
| Surface waters [b] | <0.25 | 12.48 (9.67 - 15.29) | <0.001 | 1.74 (1.48 - 2.00) | <0.001 | 0.662 | 43 |

[a]    95% confidence intervals of slopes and intercepts are in parentheses; "p" is the probability value of the slope or intercept being equal to 0 (i.e., $p < 0.05$ indicates significant difference from zero); Alk = alkalinity.

[b]    None of the surface waters had an $NO_3^-$ concentration >0.25 mmol L$^{-1}$.



**Table 3:** Averaged nitrogen budget for agricultural soils (10'490 km$^2$) in Switzerland. (FOAG, 2018).

| Input/output | annual N flux<br>kg N ha$^{-1}$ a$^{-1}$ | reference |
|---|---|---|
| **Total N supply** | **233** | FSO (2014) |
| **Fertilizer supply** | **206** | FSO (2014) |
| Manure | 124 | FSO (2014) |
| Leguminosae | 32 | FSO (2014) |
| Org/inorg commercial fertilizer | 50 | FSO (2014) |
| $NH_4NO_3$ (60%) | 30 | FOAG (2016) |
| Urea (20%) | 10 | FOAG (2016) |
| Other (20%) | 10 | FOAG (2016) |
| **N deposition with seeds** | **1** | FSO (2014) |
| **Atmospheric N deposition** | **26** | FSO (2014) |
| Atm. $NO_3^-$ deposition | 8.7 | |
| Atm. $NH_4^+$ deposition | 17.3 | |
| **Nitrogen removal with crop** | **148** | FSO (2014) |
| **Excess nitrogen deposition** | **85** | FSO (2014) |
| N removal by groundwater to surface waters | 22 | Hürdler et al. (2015a) |
| N load to the atmosphere by denitrification to N$_2$ and/or N$_2$O | 63 | |





**Figure 1:**



**Figure 1:** Relationships between $NO_3^-$ concentration and (a and b) alkalinity, (c) $\Omega_{CO_2}$ ($CO_2$ saturation index), (d) pH, (e and

f) molar $([Ca]+[Mg])/[HCO_3^-]$ ratio, and (g) $\Omega_{calcite}$ (calcite $[CaCO_3]$ saturation index) in aquifers and surface waters in calcareous catchments in Switzerland that had alkalinity $\geq 1$ mmol $L^{-1}$. These include 42 groundwaters across the country (orange), 120 wells in the Canton of Zürich (blue), 21 lakes (green), and 22 stations on 12 rivers (purple) (Table 1). The light

5   blue circle depicts the resulting alkalinity for the estimated average loads of 1.52 kmol N $ha^{-1}$ $a^{-1}$, 2.36 kmol $H^+$ $ha^{-1}$ $a^{-1}$ and 94.1 kmol $CO_2$ $ha^{-1}$ $a^{-1}$ (see Synthesis section) and a groundwater formation rate of 5000 $m^3$ $ha^{-1}$ $a^{-1}$. The solid red curve is the ChemEQL-predicted relationship assuming the $NO_3^-$ : $CO_2$ : $H^+$ ratio of the above loads in a system open to the air in the soil in the presence of calcite. The light blue square depicts the alkalinity and $NO_3^-$ concentration in a natural system without fertilization and without crop harvest, but with the current atmospheric N deposition of 1.86 kmol $ha^{-1}$ $a^{-1}$. The speciation

10   calculation was performed considering the activities of the ions (Debye-Hückel-approximation), which was essential for the appropriate estimation of the dissolution of calcite. Note the difference in scale for the horizontal axis between panels b) and f) and the other five panels. Linear-regression slopes and intercepts shown by the black lines in panels a) and b) are listed in Table 2.





Figure 2:

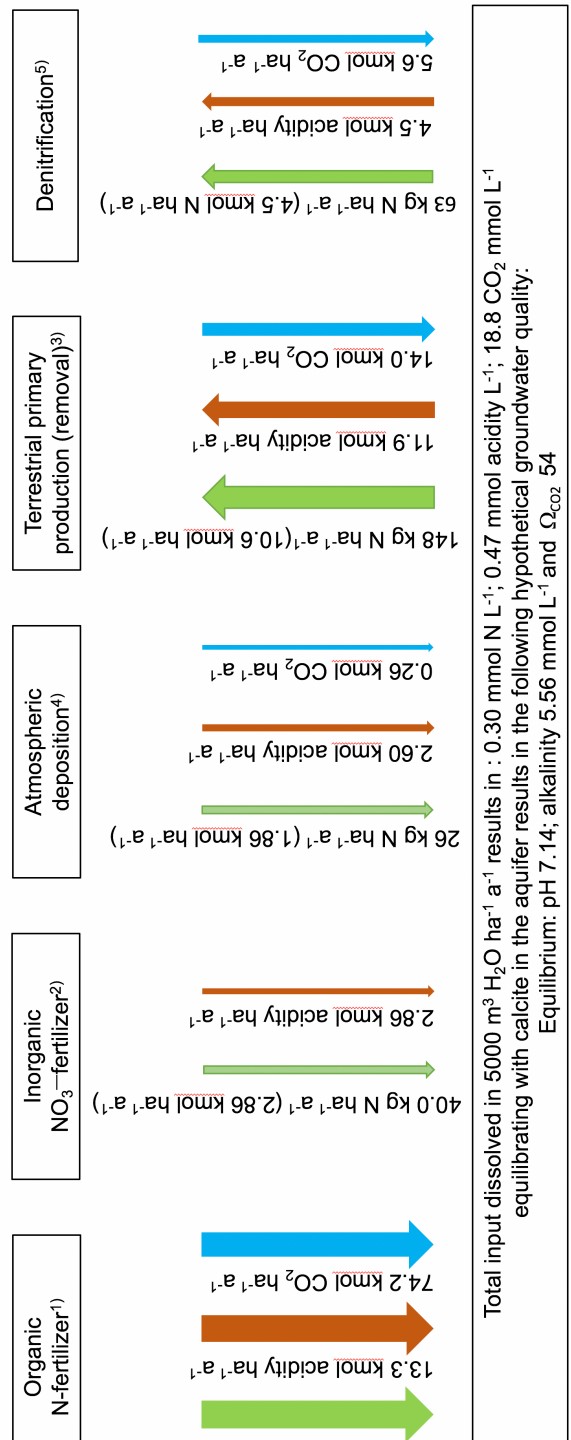

1) Mineralization of organic N-fertilizer generates $106 CO_2$ and $18$ $H^+$ per $16$ $NO_3^-$ (eq. S-9)
2) Nitrification of $NH_4NO_3$ generates $2H^+$ and $NO_3^-$ per $mol$ $NH_4NO_3$ (eq. S-2))
3) Primary production (annual crop – annual seed) i) consumes $18$ $H^+$ and ii) generates $21.2$ $CO_2$ (due to root respiration) per $16$ $NO_3^-$ removal (eq. S-9)
4) At a $pCO_2$ of $4 \times 10^{-4}$ atm, $10^4$ $m^3$ $ha^{-1}$ $a^{-1}$ rain water saturated with $CO_2$ import $0.26$ kmol $CO_2$ $ha^{-1}a^{-1}$.
   About $1/3$ of the atmospheric N-deposition is $NO_3^-$ and $2/3$ is $NH_4^+$, which is nitrified and generates $1$ equivalent of acidity per mol of oxidized $NH_3$ (eq. S-1).
   Hence, $1.34$ kmol $H^+$ $ha^{-1}a^{-1}$ originates from deposition of strong acids, $1.21$ kmol $H^+$ $ha^{-1}a^{-1}$ from nitrification of deposited $NH_4^+$.
5) According to $1.25$ $CH_2O$ + $NO_3^-$ + $H^+$ = $1.25$ $CO_2$ + $0.5$ $N_2$ + $1.75$ $H_2O$ denitrification of $1$ mol $NO_3^-$ generates $1.25$ mol $CO_2$ and consumes $1$ mol $H^+$

**Figure 2:** Compilation and overview of areal deposition rates of nitrogen, acidity, and $CO_2$ in agricultural soils of Switzerland, interacting in the carbonaceous soil- groundwater system.
Concentrations were estimated assuming a groundwater formation rate of $5000$ $m^3$ $ha^{-1}$ $a^{-1}$ .





Figure 3:

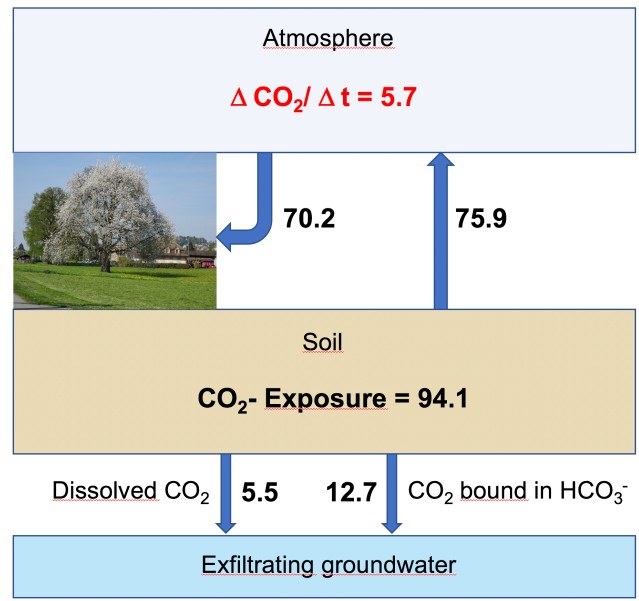

**Figure 3:** $CO_2$ budget of Swiss agricultural soils. Fluxes in kmol $CO_2$ ha$^{-1}$ a$^{-1}$.