# Peer review of "Alkalinity and nitrate concentrations in calcareous watersheds: Are they linked, and is there an upper limit to alkalinity?"

_Biogeosciences, 2018_

## Referee Comment (RC1) · Anonymous Referee #1 · 7 Dec 2018

To start with, I don't understand the title. Of course alkalinity is linked to NO3(and many other parameters too). And, of course, there is an upper limit which is the solubility product.

The authors tried to link alkalinity with NO3 which is OK in open systems albeit ignoring PO4 leads to an systematic error of 7%(e.g., Chen et al., Geochemical J. 16,1,1982). In closed systems such as groundwater or soil water the problem is huge as NO2 and NH4 come into play. The biggest issue, however, is sulfate reduction which is orders of magnitude larger that the effect of NO3(see Chen, Deep-Sea Research II, 49,5365,2002). The authors are encouraged to present dissolved oxygen data should

they wish to resubmit so that the readers get a sense of the possible roles of NO2, NH4, and SO4 etc.

Now the model. Eq 5 is valid only for phytoplankton, and is not a generic formula for organic matter as the authors claimed. Land plants have a much higher C/N ratio so the model is invalid. Should the authors wish to resubmit they should also try error analysis so that readers know how much uncertainty there is and how errors propagate.

Minor issues: 1. How is pH defined and in what scale? 2. Explain "CO2 bound in HCO3" in the caption for Fig. 3. 3. The groundwater has a pH range of 7.1-7.8 so I question the average value of 7.14 used for the model

---

## Author Comment (AC1) · 17 Dec 2018

We agree that the title is not very straight-forward and requires some interpretation. We will change the title to "Nitrogen fertilization of soils fuels carbonate weathering in calcareous watersheds". When we noticed the stunning correlation between alkalinity and nitrate concentrations in groundwaters, lakes and rivers, and the asymptotic approach to a maximum groundwater alkalinity at nitrate concentrations exceeding 0.25 mmol L-1, it was not at all evident what the linking processes were. No such correlation exists for other anions such as sulfate or nitrite. Ammonium was absent in virtually all samples, because all ground- and surface waters in our dataset contained measurable

oxygen. (We will add this information to the Results chapter, end of the 1st paragraph). Thus, we have no reason to expect the reduction of sulfate, iron or manganese etc. to play a significant role in the soil system. This is also supported by the observation that the ratio of calcium + magnesium vs. alkalinity was close to 0.5, indicating the origin of alkalinity was mainly the dissolution of Ca/Mg-carbonate minerals. The main effect of the groundwater (or hypolimnetic lake water) being "closed" to the atmosphere (i.e. not in equilibrium with the atmospheric $CO_2$ partial pressure) was that the partial pressure increased due to aerobic mineralization of organic matter, which subsequently decreased pH and thus affected the dissolution of carbonate minerals. The solubility product of calcite indeed appears to determine the concentrations of calcium ions and carbonate ions ($CO_3^{2-}$) in the groundwaters as is shown in Figure 1g and commented on lines 10-11 of page 4. Lake and river waters, however, are often supersaturated with respect to calcite (Küchler-Krischun and Kleiner, Aquatic Sci, 52, 176-197, 1990; Müller et al., Limnol. Oceanogr. 61, 341-352, 2016) for reasons that are still debated (e.g., inhomogeneities initiating the formation of initial nuclei (Obst et al., Geobiol. 7, 324-347, 2009), retardation by phosphate sorption (Giannimaras and Koutsoukos, J. Coll. Intf. Sci. 166, 423-430, 1987)). Moreover, the solubility of calcite is not a fixed upper limit to alkalinity. Instead, that upper limit increases as pH decreases, explaining in part why increases in $H^+$ and $CO_2$ as agricultural fertilization is increased can increase the alkalinity in a calcareous watershed, as we have documented in Swiss soils.

The model: We agree that the generic Redfield ratio for phytoplankton (C:N:P = 106:16:1) is not strictly applicable for land plants, because they have a higher C:N ratio. A generally applicable value is difficult to determine, however. An alternative attempt could be the use of the stoichiometry for "soil" suggested by Cleveland & Liptzin (Biogeochem. 85, 235-252, 2007) (C:N:P = 186:13:1). We currently screen the literature for element ratios and will redo our calculations more convincingly in the revised version of the manuscript. However, our main aim is to demonstrate the link between increased agricultural fertilization and increased alkalinity with a conceptual model in which the precise C:N ratio would not significantly alter our line of argument, for the

following reasons: • The numbers in Table 3, which Figure 2 is based on, reflect the Swiss nitrogen budget. It does not contain assumptions on the coupling between the CO2 and nitrogen cycles. • In Figure 2, a correction towards a higher C:N ratio in the manure would increase the CO2 production due to mineralization in the item "Organic N-Fertilizer" as would a higher C:N crop plant ratio in the item "Terrestrial primary production", because root respiration has been shown to increase in proportion to CO2 assimilation. Hence, at the given nitrogen budget, higher C:N ratios would even further increase the dissolution of carbonates driven by agricultural activities.

Minor issues: 1. How is pH defined and in what scale? pH is defined as -log(H+ activity) according to IUPAC convention. The scale is standard units.

2. Explain "CO2 bound in HCO3" in the caption for Fig. 3. Calcite dissolves using 1 equivalent of CO2 and H2O to produce Ca2+ and 2 HCO3-. Hence, 0.5 equivalents of CO2 were consumed per each equivalent of dissolved HCO3- (thus being incorporated into HCO3-). We will explain this term in the figure caption.

3. The groundwater has a pH range of 7.1-7.8 so I question the average value of 7.14 used for the model Minimum and maximum pH values in the groundwater dataset are 7.05 and 7.83, respectively, and the median pH is 7.22. The pH calculated by the speciation equilibrium program ChemEQL (chapter 4.1.7, 1st paragraph) using the estimated concentrations of H+, CO2 in equilibrium with calcite results in a pH of 7.14 (blue circle in Figure 1d), which we think agrees quite well with the measured values.
* * *

---

## Referee Comment (RC2) · Anonymous Referee #2 · 10 Jan 2019

**Review on Müller et al. (2018, BGD): Alkalinity and nitrate concentrations in calcareous watersheds: Are they linked, and is there an upper limit to alkalinity?**

Müller et al. find a covariation of alkalinity with nitrate concentration ($[NO_3^-]$) in 'aquifers in calcareous watersheds in Switzerland'. For $[NO_3^-]$ up to 0.25 mmol $L^{-1}$ alkalinity increases linearly with nitrate, for higher concentrations is levels off never exceeding 8 mmol $L^{-1}$. The authors try to explain these variations and the existence of a maximum alkalinity concentration (at 8 mmol $L^{-1}$) by various processes.

In a paper where alkalinity is a central concept (and the first word in the title) I would expect a definition or at least a reference to the definition (I suggest citing Dickson, 1981, who gave the most precise definition) and a description or reference how alkalinity (better total alkalinity, TA) was measured or estimated (for example, Dickson et al., 2007).

Based on the TA definition by Dickson (1981) and the electro-neutrality of aquatic solutions, Wolf-Gladrow et al. (2007) derived a different way to express TA in seawater, namely:

$$\begin{aligned}
& [Na^+] + 2\,[Mg^{2+}] + 2\,[Ca^{2+}] + [K^+] + 2[Sr^{2+}] + ... \\
& -[Cl^-] - [Br^-] - [NO_3^-] - ... \\
& -TPO_4 + TNH_3 - 2\,TSO_4 - THF - THNO_2 \\
& = TA^{(ec)}
\end{aligned} \quad (1)$$

where $TPO_4 = [H_3PO_4] + [H_2PO_4^-] + [HPO_4^{2-}] + [PO_4^{3-}]$, $TNH_3 = [NH_3] + [NH_4^+]$, $TSO_4 = [SO_4^{2-}] + [HSO_4^-]$, $THF = [F^-] + [HF]$, and $THNO_2 = [NO_2^-] + [HNO_2]$ are total phosphate, ammonia, sulphate, fluoride, and nitrite, respectively.

In the current context (freshwater with $Ca^{2+}$, $Mg^{2+}$, and $NO_3^-$) this expression can be simplified to

$$TA^{(ec)} \approx 2\,[Ca^{2+}] + 2\,[Mg^{2+}] - [NO_3^-] \quad (2)$$

(the other terms are probably small or roughly cancel each other). This expression for TA shows that addition of nitrate would decrease TA. The observed increase of TA with increasing nitrate is, therefore, not a direct effect of nitrate addition (concentration too small and wrong sign in the TA expression), but is rather a proxy for other processes, namely $CaCO_3$ dissolution (enhanced weathering) caused by agriculture. Although TA varies almost linearly with nitrate concentration for nitrate concentrations up to 0.25 mmol $L^{-1}$, the relation become nonlinear (levels off, saturates) for higher concentrations. This also speaks against a direct impact of nitrate, but suggests that nitrate could be a proxy for other processes.

p. 3 L2-3: "If a titrated alkalinity was not reported, we approximated it as the reported molar concentration of $HCO_3^-$."
It is not clear which values are based on proper titration and which on $HCO_3^-$ concentration. Also missing: description of measurement procedures for alkalinity, $HCO_3^-$, and $[Ca^{2+}] + [Mg^{2+}]$ (p. 4 L8: 'Ranging from 0.51 to 0.72, the molar $([Ca]+[Mg]) : [HCO_3^-]$ ratio ...')

p. 5 The use of Redfield ratios (C:N = 106:16 mol $mol^{-1}$) for land plants is unrealistic and has already been criticized by Reviewer # 1.

The impact on alkalinity by adding various nitrogen compounds (nitrate, ammonia, urea) and by conversion processes needs more attention (see Wolf-Gladrow et al. 2007 for some hints).

The authors have compiled a lot of data that are somewhat hidden in various archives. It would be great if these data could be made publicly/more easily available at, for example, the Carbon Dioxide Information Analysis Center, https://cdiac.ess-dive.lbl.gov.

Minor points:

p.2 L9-12: "Therefore, in calcareous soils, in-soil production of $CO_2$ (e.g., due to root respiration and heterotrophic mineralization of organic matter) ... result in an increased alkalinity concentration. In contrast, in the absence of carbonate minerals, alkalinity is expected to decrease in proportion to the amount ... in-soil $CO_2$ production (Perrin et al., 2008)."
This can be misleading or is wrong. Addition of $CO_2$ does not change total alkalinity. However, addition of $CO_2$ will decrease pH and may lead to dissolution of $CaCO_3$ resulting in the increase of total alkalinity.

p.2 L38-39: I suggest changing '$10^{-3.5}$ to $10^{-3.4}$ atm' to '316 to 398 $\mu$atm' (or after 'rounding': 300 to 400 $\mu$atm)

p.2 L39-41 "... in the absence of acids other than $H_2CO_3$, an alkalinity concentration of 1.42 mmol $L^{-1}$ and a pH of 8.24 would be expected in water draining a hypothetically N-free (and thus sterile) calcareous soil, assuming a groundwater temperature of 8°C." Which assumptions have been made? (I guess $\Omega_{calcite} = 1$, equilibrium of $CO_2$ partial pressures, alkalinity $\approx$ [$HCO_3$] + 2 [$CO_3^{2+}$] $\approx$ Ca) Can you give a reference? Which equilibrium constants did you use?

p.3 L12-13: "We calculated the $CO_2$ saturation index of water as $\Omega_{CO_2} = CO_2(aq)/CO_2(atm)$, where $CO_2(aq)$ is the partial pressure of $CO_2$ in the water (in atm) and $CO_2(atm)$ is the partial pressure of $CO_2$ in the atmosphere ..."
I suggest to use the notation $pCO_2$ for partial pressures.

p. 6: "As $pCO_2$ increases and as pH concurrently decreases, the extent to which $H_2CO_3$ dissociates into $HCO_3^-$ and $CO_3^{2-}$ decreases (because increasingly greater percentages of the DIC remain as $H_2CO_3$ as the $H^+$ concentration increases)."
This could be quantified and illustrated by a Bjerrum plot.

Supplement:

S-4: "p" is the probability value of the slope or intercept being equal to 0 (i.e., p < 0.05 indicates significant difference from zero)
No! $p$ is the probability to obtain an estimate $\hat{\beta}$ or more extreme values for the slope, $\beta$, under the null hypothesis $H_0$ : 'slope $\beta$ is zero'. The null hypothesis is rejected if $p < \alpha$ where $\alpha$ is the level of significance (commonly chosen as $\alpha = 0.05$). Same for the intercept.

**References**

[1] Dickson, A.G. An exact definition of total alkalinity and a procedure for the estimation of alkalinity and total inorganic carbon from titration data. *Deep Sea Research Part A. Oceanographic Research Papers*, 28(6):609–623, 1981.

[2] Dickson, A.G., C.L. Sabine, and J.R. Christian. *Guide to best practices for ocean* $CO_2$ *measurements*. North Pacific Marine Science Organization, 2007.

[3] Wolf-Gladrow, D.A., R.E. Zeebe, C. Klaas, A. Körtzinger, and A.G. Dickson. Total alkalinity: The explicit conservative expression and its application to biogeochemical processes. *Marine Chemistry*, 106(1):287–300, 2007.

---

## Author Comment (AC2) · 20 Jan 2019

Review on MuÌĹller et al. (2018, BGD): Alkalinity and nitrate concentrations in calcareous watersheds: Are they linked, and is there an upper limit to alkalinity? MuÌĹller et al. find a covariation of alkalinity with nitrate concentration ([NO3-]) in 'aquifers in calcareous watersheds in Switzerland'. For [NO3-] up to 0.25 mmol L-1 alkalinity increases linearly with nitrate, for higher concentrations is levels off never exceeding 8 mmol L-1. The authors try to explain these variations and the existence of a maximum alkalinity concentration (at 8 mmol L-1) by various processes. In a paper where alkalinity is a central concept (and the first word in the title) I would expect a definition or at least a

reference to the definition (I suggest citing Dickson, 1981, who gave the most precise definition) and a description or reference how alkalinity (better total alkalinity, TA) was measured or estimated (for example, Dickson et al., 2007).

RESPONSE: Because our manuscript deals with groundwater (not seawater) with alkalinity values exceeding 1 mmol L-1 (p. 3, line 2), we refer to Stumm and Morgan (1996) and define alkalinity as described in eq. 1 (line 35): Alkalinity [mmol L-1] = [HCO3-] + 2[CO32-] + [OH-] − [H+] (1).

We are aware of the alternative definition, which Stumm and Morgan discuss as well. However, due to the dissolution of calcite in these calcareous Swiss watersheds, alkalinity is always much higher than other acid anions. Thus, we do not see an advantage to using the alternative definition of alkalinity, and it would not change our general line of thought. Because these systems are well-buffered with respect to pH (pH 7 to 8.3, see Figure 1d in the manuscript), the concentrations of OH- and H+ are always negligible in comparison to HCO3- concentrations; and at pH 8.3, [CO32-] contributes only 2% to the alkalinity. Therefore, Alkalinity [mmol L-1] ≈ [HCO3-]. (2) In cases in which all alkalinity originated from the dissolution of carbonates, equivalent concentrations of alkalinity and the sum of Ca2+ and Mg2+ must be equal: Alkalinity [mmol L-1] = [HCO3-] = 2([Ca2+] + [Mg2+]). (3) This relationship is very helpful because the ratio ([Ca2+] + [Mg2+])/[HCO3-] (4) indicates whether alkalinity originates solely from the dissolution of carbonates by CO2 (=0.5). If part of the alkalinity originates from proton-(Ca/Mg)-carbonate interactions, the ratio exceeds 0.5. ——————

Based on the TA definition by Dickson (1981) and the electro-neutrality of aquatic solutions, Wolf-Gladrow et al. (2007) derived a different way to express TA in seawater, namely: [Na+] + 2[Mg2+] + 2[Ca2+] + [K+] + 2[Sr2+] + ... −[Cl-] − [Br-] − [NO3-] − ... −TPO4 + TNH3 − 2TSO4 − THF − THNO2 = TA(ec) (1) where TPO4 = [H3PO4] + [H2PO4-] + [HPO42-] + [PO43-], TNH3 = [NH3] + [NH4+], TSO4 = [SO42-] + [HSO4-], THF = [F-] + [HF], and THNO2 = [NO2] + [HNO2] are total phosphate, ammonia, sulphate, fluoride, and nitrite, respectively. In the current context (freshwater with Ca2+,

Mg2+, and NO3-) this expression can be simplified to TA(ec) $\approx$ 2 [Ca2+] + 2 [Mg2+] − [NO3-] (2) (the other terms are probably small or roughly cancel each other). This expression for TA shows that addition of nitrate would decrease TA. The observed increase of TA with increasing nitrate is, therefore, not a direct effect of nitrate addition (concentration too small and wrong sign in the TA expression), but is rather a proxy for other processes, namely CaCO3 dissolution (enhanced weathering) caused by agriculture. Although TA varies almost linearly with nitrate concentration for nitrate concentrations up to 0.25 mmol L-1, the relation become nonlinear (levels off, saturates) for higher concentrations. This also speaks against a direct impact of nitrate, but suggests that nitrate could be a proxy for other processes.

RESPONSE: We agree with the reviewer's perception of the subject. None of the concentrations of the accompanying base cations (except for Ca and Mg) or acid anions was high enough to bias alkalinity. In the manuscript, we hypothesized that the covariation of alkalinity and nitrate is a result of (i) mineralization of organic (nitrogen) fertilizer, (ii) exchange of OH- or H+ ions from plant roots during the uptake of NO3- or NH4+, respectively, and (iii) CO2 release (and subsequent dissolution of soil carbonates) due to fertilizer-stimulated plant growth. Thus, we do not need believe additional discussion is needed. ———————-

p. 3 L2-3: "If a titrated alkalinity was not reported, we approximated it as the reported molar concentration of HCO3-." It is not clear which values are based on proper titration and which on HCO3- concentration. Also missing: description of measurement procedures for alkalinity, HCO3-, and [Ca2+] + [Mg2+] (p. 4 L8: 'Ranging from 0.51 to 0.72, the molar ([Ca]+[Mg]) : [HCO3-] ratio ...')

RESPONSE: The Swiss Federal Office for the Environment (responsible for the NAQUA dataset) informed us that their term "HCO3-" was used instead of "alkalinity", but the parameter was determined by endpoint titration to pH 4.2, which is commonly used to measure alkalinity in groundwaters. Therefore, our remark (p. 3, line 2-3) that we replaced missing 'alkalinity' by "HCO3-" concentration is unnecessary. We will

delete this sentence in the revised version. ———————-

p. 5 The use of Redfield ratios (C:N = 106:16 mol mol-1) for land plants is unrealistic and has already been criticized by Reviewer # 1.

RESPONSE: We are currently searching for more appropriate of C:N ratios for crop plants. In the revised manuscript, we will discuss the effect of this ratio on alkalinity formation. ———————

The impact on alkalinity by adding various nitrogen compounds (nitrate, ammonia, urea) and by conversion processes needs more attention (see Wolf-Gladrow et al. 2007 for some hints).

RESPONSE: We agree that it is essential for the subject of soil acidification that conversion processes of the various nitrogen compounds applied to the soil are considered. Therefore, we placed Table 1 in the supplemental information detailing 24 pertinent chemical-transformation reactions that can be expected to generate alkalinity or acidity in soils. We refer to those reactions at many places in the manuscript, e.g. p.1, line 27-28; p.2, line 23; p.3, line 4; p.4, lines 24-26, etc. Due to the diversity of processes affecting the acid-base chemistry of fertilized soils and plant growth and in an effort to not clutter the main text, we prefer to separate this subject from the main manuscript and place it in the supplemental file. ———————-

The authors have compiled a lot of data that are somewhat hidden in various archives. It would be great if these data could be made publicly/more easily available at, for example, the Carbon Dioxide Information Analysis Center, https://cdiac.ess-dive.lbl.gov.

RESPONSE: Data sources are listed in Table 1. In addition, we will deposit the data in a FAIR-aligned data repository. ———————

Minor points: p.2 L9-12: "Therefore, in calcareous soils, in-soil production of CO2 (e.g., due to root respiration and heterotrophic mineralization of organic matter) ... result in an increased alkalinity concentration. In contrast, in the absence of carbonate minerals,

alkalinity is expected to decrease in proportion to the amount ... in-soil CO2 production (Perrin et al., 2008)." This can be misleading or is wrong. Addition of CO2 does not change total alkalinity. However, addition of CO2 will decrease pH and may lead to dissolution of CaCO3 resulting in the increase of total alkalinity.

RESPONSE: We agree with the reviewer. However, if CO2 in an aqueous solution reacts with CaCO3, HCO3- ions are formed, thus, increasing alkalinity. We will clarify this in the revised manuscript as follows: "Therefore, in calcareous soils, in-soil production of CO2 (e.g., due to root respiration and heterotrophic mineralization of organic matter) and/or generation of protons by nitrification result in an increased alkalinity concentration due to the additional dissolution of carbonates." The subsequent sentence will be omitted. —————

p.2 L38-39: I suggest changing '10-3.5 to 10-3.4 atm' to '316 to 398 $\mu$atm' (or after 'rounding': 300 to 400 $\mu$atm)

RESPONSE: We change that sentence to: "At atmospheric pCO2 ranging from 10-3.5 to 10-3.4 atm (approximately 300-400 ppm; the condition from approximately 1959 to present; NOAA, 2017)". We think the "At atmospheric pCO2" at the beginning of the sentence helps avoid confusion, but the exponential form of reporting these partial pressures is more helpful if one wants to perform speciation calculations (leaving the ppm concentration in parentheses for those more comfortable with the simple form for expressing concentrations). —————

p.2 L39-41 "... in the absence of acids other than H2CO3, an alkalinity concentration of 1.42 mmol L-1 and a pH of 8.24 would be expected in water draining a hypothetically N-free (and thus sterile) calcareous soil, assuming a groundwater temperature of 8oC." Which assumptions have been made? (I guess $\Omega$calcite = 1, equilibrium of CO2 partial pressures, alkalinity $\approx$ [HCO3-] + 2 [CO32-] $\approx$ Ca) Can you give a reference? Which equilibrium constants did you use?

RESPONSE: The calculation was performed with the chemical speciation software

ChemEQL (referenced on p. 11, lines 14-16, free download). We assumed equilibrium with calcite and atmospheric pCO2. Alkalinity was not approximated but was calculated according to eqs. 1 to 3 in the manuscript. However, we have now noticed that we did not report all the equations considered in the calculation, and the equilibrium constants used. We will report them in the revised version of the supplemental file (SI Table S3). The text in the manuscript will be extended accordingly. —————

p.3 L12-13: "We calculated the CO2 saturation index of water as $\Omega$CO2 = CO2 (aq)/CO2 (atm), where CO2 (aq) is the partial pressure of CO2 in the water (in atm) and CO2 (atm) is the partial pressure of CO2 in the atmosphere ..." I suggest to use the notation pCO2 for partial pressures.

RESPONSE: We will change the expression. —————

p.6: "As pCO2 increases and as pH concurrently decreases, the extent to which H2CO3 dissociates into HCO3- and CO32- decreases (because increasingly greater percentages of the DIC remain as H2CO3 as the H+ concentration increases)." This could be quantified and illustrated by a Bjerrum plot.

RESPONSE: We will extend the sentence indicating that this process is best illustrated by a Bjerrum plot and refer the reader to Stumm and Morgan (1996, p. 160). —————————-

Supplement: S-4: "p" is the probability value of the slope or intercept being equal to 0 (i.e., p < 0.05 indicates significant difference from zero) No! p is the probability to obtain an estimate $\beta\ddot{E}$ or more extreme values for the slope, $\beta$, under the null hypothesis H0 : 'slope $\beta$ is zero'. The null hypothesis is rejected if p < $\alpha$ where $\alpha$ is the level of significance (commonly chosen as $\alpha$ = 0.05). Same for the intercept.

RESPONSE: The reviewer is correct; and that incorrect wording also appears in footnote a in Table 2. In both places, in order to avoid extensive rewording, we will revise the incorrectly-worded clause to the following: "p" is the Type I error probability for a

test of the null hypothesis that the slope (or intercept) equals 0. ———————

---

## Author Comment (AC3) · 11 Feb 2019

Concerning the request to publish the data in a public databank, we think that we are not eligible to do so as we are not the owners of the data. Owners of the groundwater data are the authorities of the Canton of Zürich and the Swiss Federal Office for the Environment. They will provide the data on request as we have stated in Table 1. Data from the River Monitoring Program (NADUF) are readily downloadable from the internet site given in the references. In any case, we are ready to share all data used in the manuscript with anyone interested in it.